# Development of a Citric-Acid-Modified Cellulose Adsorbent Derived from *Moringa peregrina* Leaf for Adsorptive Removal of Citalopram HBr in Aqueous Solutions

**DOI:** 10.3390/ph15060760

**Published:** 2022-06-17

**Authors:** Syed Najmul Hejaz Azmi, Wafa Mustafa Al Lawati, Umaima Hamed Abdullah Al Hoqani, Ekhlas Al Aufi, Khalsa Al Hatmi, Jumana Salim Al Zadjali, Nafisur Rahman, Mohd Nasir, Habibur Rahman, Shah A. Khan

**Affiliations:** 1Applied Sciences Department (Chemistry Section), Higher College of Technology, University of Technology and Applied Sciences, Al-Khuwair 133, Muscat P.O. Box 74, Oman; wafamustafa@hct.edu.om (W.M.A.L.); ekhlasalaufi@gmail.com (E.A.A.); kalhatmi94@gmail.com (K.A.H.); uman9555@gmail.com (J.S.A.Z.); 2Applied Sciences Department (Biology Section), Higher College of Technology, University of Technology and Applied Sciences, Al-Khuwair 133, Muscat P.O. Box 74, Oman; umaima.alhoqani@hct.edu.om; 3Department of Chemistry, Aligarh Muslim University, Aligarh 202002, Uttar Pradesh, India; nafisurrahman05@gmail.com (N.R.); nasirkhanazmi@gmail.com (M.N.); 4Department of General Studies, Jubail Industrial College, P.O. Box 10099, Jubail Industrial City 31961, Saudi Arabia; habibur_r@jic.edu.sa; 5Department of Pharmaceutical Chemistry, College of Pharmacy, National University of Science and Technology, PC 130, Muscat P.O. Box 620, Oman; shahalam@nu.edu.om

**Keywords:** *Moringa peregrina* leaf, citric acid, adsorbent, citalopram HBr, Langmuir isotherm, Freundlich isotherm

## Abstract

A citric-acid-modified *Moringa peregrina* leaf substrate was prepared and studied as an effective adsorbent for the adsorptive removal of citalopram HBr (CTM). FTIR spectra were utilized to characterize the prepared solid. The effects of experimental variables on the percentage removal of citalopram HBr were investigated using response surface methodology. The optimum conditions selected for removal of CTM were 7 and 4 min, 0.17 g per 50 mL and 35 mg·L^−1^ for pH, contact time, adsorbent dose and initial concentration of CTM, respectively. Under the optimized experimental conditions, 82.59% CTM (35 mg·L^−1^) was removed. The Langmuir isotherm, Freundlich isotherm, pseudo second-order kinetic model and diffusion-chemisorption model explained the adsorption data successfully. The maximum adsorption capacity at 298 K was 8.58 mg·g^−1^. A thermodynamic study illustrated that CTM adsorption was spontaneous and endothermic in nature.

## 1. Introduction

The environmental impact of pharmaceutical industries was considered to be insignificant in the early stages of drug manufacturing. But the discovery of pharmaceuticals in surface waters from 1994 onwards led to a review of the said hypothesis [1]. The hazardous effects of pharmaceutical residues in the environment have been the subject of emerging research in recent years [2,3,4]. Pharmaceuticals, such as antidepressants along with selective serotonin reuptake inhibitors (SSRIs), have received extensive attention because they are frequently discharged into water bodies through treated wastewater effluent [5,6,7]. SSRIs were among the most frequently prescribed pharmaceuticals in the United States during 2007–2008 for treatment of adolescents aged 12–18 years and were on the top of prescribed drugs for adults aged 20–59 years [8]. In Canada, the prescription rate of SSRIs by all specialized doctors for children and adolescents increased by 44% during 2005–2009 [9].

Citalopram HBr (i.e., (±)-1-(3-dimethylaminopropyl)-1-(4-fluorophenyl)-1,3-dihydroisobenzofuran-5-carbonitrile, hydrobromide) is an SSRI and frequently used for the treatment of anxiety, panic disorder and body dysmorphic disorder. Citalopram HBr (CTM) is the most prescribed drug in the world and, hence, causes the contamination of aquatic environments due to the fact of their release into municipal wastewater effluents [10,11]. A review was published that focused on the detection of psychoactive drugs, including CTM, in different aqueous matrices [12]. CTM has been detected in the effluents of sewage treatment plants in northern and southern India [13]. Moreover, the effluent from a wastewater treatment plant (Patancheru Enviro Tech Ltd., Hyderabad, India) contained a high concentration of CTM (430 µg·L^−1^) [14]. The occurrence of antidepressants, including CTM, in the rivers of Madrid (Spain) was attributed to the discharge of sewage treatments plants [15]. The occurrence of CTM in the influents and effluents of six wastewater treatment plants in Hong Kong has been reported [16]. SSRIs were only partly removed from freshwaters, and their low concentrations affect the behavior of freshwater organisms in natural habitats. CTM affects the behavior of marbled crayfish even at 1 µg·L^−1^ [17]. The environmental pollution attributed to antidepressants will increase in the coming years due to the COVID-19 pandemic which has added to the incident of anxiety and depression. This leads to more consumption of antidepressant drugs and, subsequently, causes elevated levels of these drugs in aquatic environments [18]. Therefore, it is required to develop methods for removal of CTM from aquatic environments.

In order to remove CTM from aquatic environments, various water treatment technologies, such as oxidation [19], photocatalytic degradation [20], nanofiltration [21], aerobic and anaerobic treatments [22,23] and adsorption [24], have been proposed. Among these treatment technologies, adsorption is considered a promising technique for removing pharmaceutical micropollutants from wastewater due to the availability of a variety of adsorbents and their low-cost, reusable and flexible operation. Primary paper mill sludge was pyrolyzed at 800 °C for 2.5 h to produce the adsorbent that was tested to remove CTM from water [25]. Magnetic nanoparticles modified with sodium dodecyl sulphate were used effectively to extract CTM from wastewater and estimated spectrophotometrically at 239 nm [26]. Masud and coworkers [27] synthesized a reduced graphene oxide/zero-valent iron nanohybrid that showed more than a 95% removal efficiency of CTM from water. The adsorption behavior of CTM onto porous alumina coated with natural zeolite was investigated in a variety of water samples. The elimination performance varied between 75% and 84% [28]. Zhiteneva et al. developed biologically activated carbon filters and used them to eliminate trace organic chemicals, including CTM, from water [29].

The adsorption process involved interactions among operating variables in a nonlinear way. In such cases, the conventional method (one variable at a time method) for optimization of the adsorption process is not effective, because it does not consider the interaction effects among the operating variables. Additionally, the conventional method requires large numbers of experimental runs and, thus, is time consuming and requires more chemicals to complete the optimization process. To overcome these shortcomings, the design of experiments and a response surface methodology (RSM) have been found to be highly effective for developing, improving and optimizing the adsorption processes [30,31,32]. Moreover, RSM considers the interaction among the operating variables with a limited number of experimental runs when determining the optimum operating conditions of the processes.

According to a literature review and gathered information, there is no report regarding the use of chemically modified biomass material of *Moringa peregrina* leaf as an adsorbent for the removal of CTM from aqueous solutions. The aim of this research was to synthesize chemically modified *Moringa peregrina* leaf adsorbent and then apply it for the removal of CTM from aqueous solutions by batch adsorption methods. Adsorption data were investigated by isotherm models. Kinetic and thermodynamic studies were also performed.

## 2. Results and Discussion

A European patent reported that when cellulose reacts with citric acid, a water-insoluble product is formed [33]. The product has excellent stability with high ion-exchange capacity and selective removal of cations (i.e., heavy metal ions, drugs and dyes). The same concept was exploited by chemists for removal of hardness in water [34], heavy metals and methylene blue [35]. Oman is rich in *Moringa peregrina* plants, and the leaf of this plant are rich in cellulose [36]. These plants are well adapted to the extreme environmental conditions of the country [37].

In this research, *Moringa peregrina* leaf biomass material was used as the source of cellulose and treated with NaOH and citric acid to form a modified bio-adsorbent, which should help to enhance the adsorption capacity of the biomass material. The treated biomass material was used to remove CTM from aqueous solutions. *Moringa peregrina* leaf biomass and *Moringa peregrina* leaf biomass treated with NaOH and citric acid have shown removal efficiencies of 59.45%, 71.36%, and 82.59%, respectively. Therefore, *Moringa peregrina* leaf biomass treated with citric acid was selected for further studies.

The FTIR spectrum of neutralized leaf adsorbent was scanned in the range of 4000–400 cm^−1^. The neutralized leaf adsorbent showed a broad band at 3422 cm^−1^ and a sharp peak at 1035 cm^−1^ for O–H and C–O stretching vibrations, respectively (Figure 1a). These peaks were due to the presence of cellulose and hemicellulose in the *Moringa peregrina* leaf adsorbent [38]. The peak at 2915 cm^−1^ was observed due to the vibrational stretching of C–H bond of alkane groups [39]. The peaks in the range from 1609 to 1535 cm^−1^ were attributed to the vibrational stretching of the C=O bond of the carboxylic group [40]. In addition to the above said bands, the band at 1700 cm^−1^ was ascribed to the carbonyl group [41]. The neutralized leaf adsorbent material was treated with a dilute NaOH solution for de-esterification, which provided stability to the adsorbent material by removing soluble substances with a low molecular weight [33]. Hemicellulose was hydrolyzed using base, whereas cellulose and lignin were resistant to hydrolysis. The FTIR spectrum of NaOH-treated neutralized adsorbent is shown in Figure 1b. The NaOH-treated neutralized adsorbent was esterified with citric acid. The FTIR spectrum of the adsorbent material treated with citric acid is shown in Figure 1c. The comparison of IR spectra of *neutralized Moringa peregrina* adsorbent (untreated with citric acid, Figure 1a) with chemically modified *Moringa peregrina* adsorbent (treated with citric acid, Figure 1c) revealed a characteristic stretching vibration absorption band at 1700 cm^−1^ attributed the presence of a carboxylic group. The broad absorption peak at 3311 cm^−1^ confirmed the existence of carboxylic O–H groups [27], hence, the reflected citric acid esterification. Based on the IR spectra (Figure 1) and literature review [42], the reaction sequence for citric-acid-modified cellulose is proposed in Figure 2.

In this study, SEM images (Figure 3a,b) were used to investigate the change in the morphological features of *Moringa peregrina* biomass treated with citric acid. An SEM image of *Moringa peregrina* biomass (Figure 3a) showed the assemblage of fine particles with void volumes. An SEM image of citric-acid-modified *Moringa peregrina* biomass (Figure 3b) showed a sheet-like structure with void volumes. This indicated that the surface was modified on treatment with citric acid.

### 2.1. Effect of pH on the Adsorption

The point of zero charge of citric-acid-modified *Moringa peregrina* biomass was evaluated and found to be 4.8, which indicated that the surface of the adsorbent was negatively charged above pH 4.8. To study the effect of solution pH on the removal efficiency, a series of experiments were performed using 35 mg·L^−1^ CTM containing solutions of varying pH values (i.e., pH 3–11). Figure 4 shows that the percent removal of CTM increased with the increasing pH of the solution and attained the maximum removal in the pH range 6.5–7.5. Above this pH, the removal efficiency decreased. The main factors which affected the removal efficiency were the CTM species and the charge on the surface of the adsorbent. The CTM exists as a cation in the pH range 5–9. Above pH 5, the surface of the adsorbent is negatively charged and, therefore, electrostatic interaction occurs between negatively charged surface and cationic species of CTM, and this interaction is maximum in the pH range of 6.5–7.5. Above pH 7.5, the hydroxyl ions concentration increases, which suppresses the electrostatic interaction between the adsorbent surface and cationic species of CTM and, thus, causes the decrease in removal efficiency.

### 2.2. Optimization of Variables

#### 2.2.1. Fitting of the Model and Relevant Statistical Analysis

The experimental results for the optimization of variables (Table 1) were fitted to linear, two-factor interaction and quadratic models. The statistical data of the polynomial models are listed in Table 2. The suitability of the model was assessed based on highest F, highest R^2^, lowest *p*-value and minimum predicted residual sum of squares (PRESS). Among these models, the quadratic model yielded the best results (highest F-value: 6.4 × 10^8^; highest R^2^: 1.0; lowest *p*-value: <0.0001; minimum PRESS value of 1951.71), hence, demonstrating the adequacy of the quadratic model. The quadratic model equation for predicting the response can be expressed as:Removal (%) = +82.59 + 2.94A + 17.87B + 13.26C + 3.61AB − 1.13AC + 7.69BC − 11A^2^ − 22.26B^2^ − 29.03C^2^(1)

The plot of the experimental values versus the predicted values (Figure 5) showed an excellent correlation with an R^2^ of 1.0000. Adequate precision estimates of the signal (S)-to-noise (N) ratio should be above four to obtain an adequate signal. Herein, the value of this ratio was 51,557.25, hence, illustrating the adequacy of the quadratic model. Moreover, the values of the adjusted R^2^ and predicted R^2^ were identical, confirming the suitability of the model.

#### 2.2.2. Effect of the Adsorption Parameters on the Responses

Three-dimensional response surface plots were plotted to examine the interactive effect of two variables on the % removal of CTM while holding the other variables at a constant level (Figure 6). The graph (Figure 6a) for the interactive effect of solution pH and contact time at a constant adsorbent dose (0.17 g) and initial concentration of citalopram (40 mg·L^−1^) revealed that at any fixed contact time, the removal efficiency increased with increasing pH, and the maximum removal efficiency was achieved at pH 7. Figure 6b shows the combined effect of contact time and adsorbent dosage while the solution pH was fixed at 7 and the initial concentration at 40 mg·L^−1^. The increase in the adsorbent dosage as well as in contact time resulted in an increase in the removal efficiency of citalopram. The maximum removal (82.59%) was achieved with an adsorbent dose of 0.17 g and a contact time of 4 min. The mutual effects of the pH and adsorbent dose are presented in Figure 6c. At any fixed adsorbent dose, the % removal increased with increasing pH, and the maximum percent removal of citalopram was obtained at pH 7.

#### 2.2.3. Numerical Optimization Using the Desirability Function

A desirability function was considered to evaluate the optimal levels of the operating variables to obtain the maximum removal efficiency. Herein, the adsorption variables, such as pH, contact time, and adsorbent dose, were set at their corresponding −1 to +1 levels and the % removal as response was considered at the highest (maximum). The Design Expert software gave many solutions, and the most appropriate solution was with a higher desirability index chosen to obtain the optimal conditions. The desirability function-based profile for optimizing the removal of citalopram is presented in Figure 7. It was observed that the maximum removal of CTM (82.59%) was realized with a pH of 7, contact time of 4 min, adsorbent dose of 0.17 g with a desirability index = 1.0. Under the optimized conditions, the experiments were executed in triplicate to evaluate the % removal of citalopram. The experimental results agreed well with those predicted by RSM, validating the optimized conditions.

### 2.3. Adsorption Isotherms

Langmuir, Freundlich, Temkin and Dubinin–Radushkevich isotherm models were applied to investigate the equilibrium adsorption data. In addition, adsorption isotherm studies were important to understand the nature of the interaction between CTM and citric-acid-treated *Moringa peregrina* leaf adsorbent.

The Langmuir isotherm model is described by the following linearized expression [43]:(2)$$\frac{1}{q_{e}}=\frac{1}{q_{m}}+\frac{1}{K_{L}q_{m}}\times \frac{1}{\mathrm{Ce}}$$
where $q_{e}=$ adsorption capacity determined at equilibrium (mg·g^−1^); q_m_ = maximum adsorption capacity in mg·g^−1^; K_L_ = Langmuir constant in L·mg^−1^. The Langmuir plot constructed between 1/q_e_ and 1/C_e_ yielded a straight-line equation of 1/q_e_ = −0.1165 + 1.4439 (1/C_e_) with coefficient of correlation (R^2^) = 0.9999 (Figure 8a). q_m_ and K_L_ values were calculated and found to be 8.58 mg·g^−1^ and 0.08 L·mg^−1^, respectively. The most characteristic property of the Langmuir isotherm is given by R_L_ (separation factor, a dimensionless quantity). The R_L_ value at the initial concentration of citalopram was calculated using the following equation:(3)$$R_{L}=\frac{1}{1+K_{L}C_{i}}$$
where C_i_ is the initial concentration of adsorbent (mg·L^−1^). The R_L_ values were in the range of 0.24–0.55 at room temperature and, hence, indicated that the adsorption of citalopram HBr onto the surface of the citric-acid-treated *Moringa peregrina* leaf adsorbent was favorable at 298 K. The findings and values obtained using the Langmuir isotherm equation demonstrated a monolayer adsorption of CTM. The literature review revealed similar results with chloramphenicol [44] and ampicillin [45] onto modified adsorbents.

The Freundlich isotherm model is described by the following linearized expression [46]:ln q_e_ = ln K_f_ + 1/n·ln C_e_(4)
where K_f_ = adsorption capacity; n = intensity of adsorption.

The Freundlich plot constructed between ln q_e_ and ln C_e_ yielded a straight line with a coefficient of correlation of R^2^ = 0.9978 (Figure 8b). The K_f_ and n values were calculated and found to be 0.51 and 0.65, respectively. The low value of n (<1) corresponded to the poor adsorption potential. The value of 1/n greater than 1 implied cooperative adsorption [47]. The literature review revealed similar results with sulfamethoxazole onto sugarcane bagasse-derived biochar [48].

The Temkin isotherm model is described by the following linearized expression [49]:q_e_ = B_T_·ln A_T_ + B_T_·ln C_e_
(5)
where B_T_ (heat of adsorption in J·mol^−1^) = RT/b_T_; A_T_ = equilibrium-binding constant of the Temkin isotherm in Lg^−1^; b_T_ = the Temkin isotherm constant; R = universal gas constant taken as 8.314 J·mol^−1^K^−1^; T = temperature taken as 298 K. The Temkin plot constructed between q_e_ and lnC_e_ yielded a straight line with a coefficient of correlation of R^2^ = 0.957 (Figure 8c). The A_T_ and B_T_ values were calculated and found to be 0.485 Lg^−1^ and 7.563 J·mol^−1^, respectively. The high value of B_T_ indicated a physical adsorption process.

The Dubinin–Radushkevich isotherm model is described by the following linearized expression [50]:ln (q_e_) = ln (q_s_) + K_ad_ ε^2^(6)
where q_e_ = amount of adsorbate adsorbed onto the adsorbent at equilibrium (mg·g^−1^); q_s_ = theoretical isotherm saturation capacity (mg·g^−1^); K_ad_ = the Dubinin–Radushkevich isotherm constant (mol^2^·J^−2^); ε = the Dubinin–Radushkevich isotherm constant.

ε was calculated using the following expression:ε = RT ln (1 + 1/C_e_)(7)

The Dubinin–Radushkevich isotherm plot constructed between ln q_e_ and ε^2^ yielded a straight-line equation with a coefficient of correlation of R^2^ = 0.962 (Figure 8d). The $q_{s}$ and $K_{\mathrm{ad}}$ values were calculated and found to be 11.00 mg·g^−1^ and −2 × 10^−6^ mol^2^·J^−2^, respectively. Mean free energy (E) per molecule of citalopram HBr was computed using the following expression [50]:(8)$$E=\frac{1}{\sqrt{2{\;K}_{\mathrm{ad}}}}$$

The E value was calculated and found to be −0.50 kJ·mol^−1^. The high value of E indicated a physisorption process.

The correlation coefficients (R^2^) for the Langmuir and Freundlich isotherm models indicated a strong relationship [51] between the opted variables, as the R^2^ values were greater than 0.99, but in the case of the Temkin and Dubinin–Radushkevich models, the R^2^ values were between 0.95 and 0.99, indicating a moderate relationship between variables. Hence, the Langmuir and Freundlich isotherm models were able to describe the adsorption process very well. In addition to the R^2^ value calculation, a statistical approach (an error function called chi-square) was adopted for comparing the goodness of fit of the adsorption isotherms using the following expression:(9)$$\chi^{2}=\overset{n}{\underset{i=0}{\sum }}{\left[\frac{{\left(q_{e,\;\exp}-q_{e,\;\mathrm{cal}}\right)}^{2}}{q_{e,\;\mathrm{cal}}}\right]}_{i}$$

The chi-square values for all opted models were calculated and found to be 3.24 × 10^−3^, 3.77 × 10^−2^, 0.65 and 1.04 for Langmuir, Freundlich, Temkin and Dubinin–Radushkevich, respectively. The highest R^2^ value and the lowest chi-square value were obtained for the Langmuir isotherm model. Hence, the Langmuir isotherm model was best suited to describe the uptake of CTM onto chemically modified *Moringa peregrina* leaf adsorbent (Table 3).

### 2.4. Adsorption Thermodynamics

The orientation and feasibility of the uptake of citalopram HBr onto chemically modified *Moringa peregrina* leaf adsorbent was judged by the Gibb’s free energy (ΔG°) at room temperature using the following equation:(10)$$\Delta G{}^{\circ}=-{\mathrm{RTlnK}}_{c}$$
where K_c_ (equilibrium constant) = q_e_ ×1000/C_e_; R = universal gas constant (8.314 JK^−1^mol^−1^); T = absolute temperature in Kelvin. ΔG° was calculated and found to be −17.89 kJ·mol^−1^. Hence, the uptake of CTM onto chemically modified *Moringa peregrina* leaf adsorbent was spontaneous, feasible and resulted in physicochemical sorption. The increment in temperature (greater than 298 kelvins) did not affect the uptake of CTM onto chemically modified *Moringa peregrina* leaf adsorbent. Hence, the sorption of CTM was endothermal in nature.

### 2.5. Adsorption Kinetic Model

To study the adsorption efficiency of CTM onto chemically modified *Moringa peregrina* leaf adsorbent, pseudo first-order and pseudo second-order kinetic models were utilized.

The linearized pseudo first-order kinetic model is illustrated as:ln (q_e_ − q_t_) = ln q_e_ − k_1_t(11)
where q_t_ = adsorbate amount in mg on to adsorbent (g) at time t (min); k_1_ = rate constant in min^−1^. Ln (q_e_ − q_t_) was plotted against t and provided a straight equation of ln (q_e_ − q_t_) = 0.9499−0.6249 t with a coefficient of correlation (0.9975) (Figure 9a). q_e_ was calculated and found to be 2.59, which was much smaller than the experimental q_e_ (8.48 mg·g^−1^). Hence, the adsorption of CTM onto citric-acid-modified *Moringa peregrina* leaf adsorbent did not obey the pseudo first-order kinetic model (Table 4).

The pseudo second-order kinetic model equation was followed, and the adsorption data was analyzed using the following equation:(12)$$\frac{t}{q_{t}}=\frac{1}{k_{2}q_{e}^{2}}+\frac{1}{q_{e}}\times t$$
where k_2_ = pseudo second-order rate constant in g mg^−1^ min^−1^.

A curve between t/q_t_ and t was plotted which provided a straight-line equation of t/q_t_ = 0.0202 + 0.1177 t with a coefficient of correlation (0.9995) (Figure 9b). q_e_ was calculated and found to be 8.50 mg·g^−1^, which was very close to the experimental q_e_ (8.48 mg·g^−1^). The high value of the correlation coefficient and the closeness of q_e_ (calculated using pseudo second-order kinetic model equation) with the experimental q_e_ confirmed the best fitting of the pseudo second-order kinetic model. Therefore, the adsorption of CTM onto citric-acid-modified *Moringa peregrina* leaf adsorbent was favored by the chemisorption process, supporting the multilayer adsorption onto the surface of the adsorbent.

### 2.6. Kinetic Adsorption Mechanism

The kinetic adsorption mechanism was analyzed by the Weber and Morris intraparticle diffusion model [52] and the diffusion-chemisorption model [53].

#### 2.6.1. Intraparticle Diffusion Model

The equation for the Weber and Morris intraparticle diffusion model is illustrated using the following equation:(13)$$q_{t\;}=C_{\mathrm{id}\;}+K_{\mathrm{id}}t^{1/2}$$
where K_id_ = intraparticle diffusion rate constant (mg·g^−1^ min^−1/2^); C_id_ = thickness of the boundary layer (mg·g^−1^), which is a constant parameter. The plot of q_t_ versus t^1/2^ yielded a linear relationship of q_t_ = 6.438 + 1.159 t^1/2^ with a coefficient of correlation of R^2^ = 0.981 (Figure 9c). The linear equation provided the intercept of 6.438 which was ˃ than zero, hence, supporting the fact that both film and intraparticle diffusion played significant roles in the adsorption of CTM.

#### 2.6.2. Diffusion-Chemisorption Model

The diffusion-chemisorption model was selected to describe the adsorption of CTM onto the citric-acid-modified *Moringa peregrina* leaf adsorbent. The equation is expressed as follows:(14)$$\frac{t^{1/2}}{q_{t}}=\frac{1}{K_{\mathrm{dc}}}+\frac{1}{q_{e}}\times t^{1/2}$$
where K_dc_ is the diffusion-chemisorption rate constant.

The plot of (t^1/2^/q_t_) versus t^1/2^ yielded a straight line with a coefficient of correlation of R^2^ = 0.9995 (Figure 9d). The curve fit well to the diffusion-chemisorption model because of an improved (greater) coefficient of correlation value (R^2^ = 0.9995) compared to the R^2^ value obtained in the intraparticle diffusion model (R^2^ = 0.981). Hence, the adsorption of the drug could be described well by the diffusion-chemisorption model. The calculated q_e_ (8.50 mg·g^−1^) obtained by the slope of the diffusion-chemisorption equation was very close to the experimental q_e_ (8.48 mg·g^−1^) (Table 4).

### 2.7. Comparison with other adsorbents

The adsorption capacity and removal efficiency of different adsorbents for CTM were compared with the developed citric-acid-modified cellulose adsorbent derived from *Moringa peregrina* leaf. Table 5 shows the characteristic properties of the adsorbents employed to remove CTM from water samples. Adsorbent prepared from pyrolysis of paper mill sludge showed an adsorption capacity of 8.50 mg·g^−1^ with a contact time of 30 min [25]. The removal efficiencies of sodium dodecyl sulphate (SDS) coated with magnetic particles [26] and different adsorbents (i.e., reduced graphene oxide (rGO), nanoscale zero valent iron (nZVI) and rGO/nZVI) [27] for CTM were found to be 79.80%, 26.30%, 31.41% and 47.90%, respectively when adsorbents were equilibrated for 10 min. Porous alumina coated with natural zeolite exhibited CTM removal efficiency in the range 75.0–84.0%, depending upon the type of water sample [28]. The adsorbent used in this study exhibited better performance with a removal efficiency of 82.59% and adsorption capacity of 8.58 mg·g^−1^ which was achieved in 4 min.

## 3. Materials and Methods

### 3.1. Apparatus

The concentration of citalopram HBr in the solution phase (before adsorption and after adsorption) was determined by measuring absorbance at 238 nm using an Evolution 300 UV–Visible spectrophotometer (Thermo Fisher Scientific, Madison, WI, USA). The SBS40 water bath thermostat (Stuart, Staffordshire, UK) was used to maintain the temperature of the solutions. The pH of the solutions was maintained with 0.10 M HCl or 0.1 M NaOH solution and measured using a Hanna pH meter (Woonsocket, RI, USA). An SM30 electrical linear plate shaker (Edmund Bühler GmbH, Bodelshausen, Germany) was used to stir solutions. A Rocker 400 electrical vacuum pump (Kaohsiung, Taiwan) was used for filtration. The unadsorbed citalopram HBr in the solution phase was filtered using a membrane filter (polyethersulfone, Auburn, AL, USA) of 0.45 µm.

Dry leaves of *Moringa peregrina* were blended using Preethi TRIO Blender (India) and sieves using 60 mesh sieve. FTIR spectra were recorded in the range of 4000–400 cm^−1^ using Fourier transform infrared spectrophotometer (IRAffinity-1, Shimadzu, Japan). A scanning electron microscope (model: JSM-7600 F, JEOL Ltd., Tokyo, Japan) was used to investigate the surface morphology of the adsorbent. An Ultrasonic sonicator (2800 series, Branson, New Carlisle, OH, USA) was used to sonicate the powdered *Moringa peregrina* leaf. OriginPro 2020b software (Northampton, MA, USA) was used to draw the spectral graphs.

### 3.2. Reagents and Standards

All reagents and solvents used were of high quality and analytical reagent grade. Citalopram HBr (100 ppm, National Pharmaceutical Company, Al-Rusyl, Oman) solution was prepared by dissolving 0.01 g citalopram HBr in 100 mL doubly distilled water. NaOH (0.025 M, Honeywell, Charlotte, NC, USA) solution was prepared in doubly distilled water. Citric acid (0.28 M, Sigma-Aldrich, Darmstadt, Germany; 192.124 g·mol^−1^) was prepared in doubly distilled water.

Leaves of *Moringa peregrina* were collected from Al-Rustaq, Al-Batinah governorate, Oman. The leaves were thoroughly rinsed with tap water and then with distilled water to remove dirt, any solid residue and particulate matter. The cleaned leaves were air-dried for 7 days in a large stainless-steel sieve. Fifty grams of dried leaves was kept in an oven at 60 °C for 24 h and milled using a blender. The powdered leaves were sieved using a 60 mesh sieve, and the sieved fine powder was transferred into an amber bottle container. The container was kept in the dark to prevent decomposition of the biomass material and used for further studies.

### 3.3. Synthesis of Chemically Modified Moringa peregrina Leaf Adsorbent

The synthesis of chemically modified *Moringa peregrina* leaf adsorbent was carried out in three steps.

Step 1: A mass of 5 g of fine *Moringa peregrina* leaf powder was taken with 200 mL distilled water (pH = 7) and stirred at 25 °C for 1 h at 300 rpm using a linear mechanical shaker. Supernatant liquid was decanted, and the residue was washed with distilled water until the pH of the residue was neutral. The neutralized residue was dried in an oven at 60 °C for 24 h and kept in a desiccator for 1 day and then weighed. The FTIR spectrum of the dried powder was recorded;

Step 2: The dried *Moringa peregrina* leaf adsorbent (neutralized) was taken in a 250 mL conical flask and mixed with 100 mL of 0.025 M NaOH. The mixture was stirred for 1 h at 300 rpm using a linear mechanical shaker at 25 °C. Supernatant liquid was decanted, and the residue was washed well with distilled water until the pH reached 7. The neutralized adsorbent was dried in an oven for 24 h at 60 °C and kept in a desiccator for one day and weighed. The FTIR spectrum of the NaOH-treated powder was scanned;

Step 3: The dried NaOH-treated adsorbent (neutralized) was taken in a 250 mL conical flask and mixed with 50 mL of 0.28 M citric acid. The mixture was stirred at 300 rpm for 30 min using a linear mechanical shaker at 25 °C. The mixture was kept in an oven for 90 min at 60 °C. The residue was washed with distilled water until the pH became 7. *Moringa peregrina* leaf adsorbent was dried in an oven at 60 °C for 24 h. The adsorbent modified with citric acid was kept in desiccator and weighed. The FTIR spectrum of the citric-acid-modified adsorbent was taken.

### 3.4. Experimental Design for Optimization

The Box–Behnken design (BBD) was applied to investigate the effect of operating variables and their interactions on the % removal of citalopram. In order to achieve the optimum conditions, three operating variables, such as contact time (A), pH (B) and adsorbent dose (C), were opted. The effect of each variable was examined at three levels (−1, 0 and +1) on the % removal of CTM. The ranges and levels of the selected factors are presented in Table 6.

The BBD matrix was obtained from Design Expert 12.0.1.0. software (trial version) and, therefore, experiments were executed according to the developed matrix in batch mode to estimate the % removal of citalopram. The adsorption data were fitted into the following second-order polynomial model [54]:(15)$$R_{C}={\;\beta }_{o}+\overset{n}{\underset{i=1}{\sum }}\beta_{i}X_{i}+\overset{n}{\underset{i=1}{\sum }}\beta_{\mathrm{ii}}X_{i}{}^{2}\overset{n-1}{\underset{i=1}{\sum }}\overset{n}{\underset{j=2}{\sum }}\beta_{\mathrm{ij}}X_{i}X_{j}+\varepsilon$$
where R_c_ = predicted removal (%); β_o_, β_i_, β_ii_ and β_ij_ = regression coefficients for intercept, linear, squared and interaction terms, respectively. X_i_ and X_j_ represent independent adsorption variables, and ε is the random error. ANOVA (i.e., analysis of variance of the quadratic model) was performed to characterize significant and insignificant terms in the quadratic model. The response surface plots (3D) were generated using second-order quadratic model for operating variables and analyzed to obtain the optimum values of adsorption variables.

### 3.5. Adsorption Procedure

Chemically modified *Moringa peregrina* leaf adsorbent (0.17 g) was transferred into a 100 mL stoppered conical flask followed by the addition of 50 mL of CTM solution (1.5–40 mg·L^−1^, pH 7). The mixture was kept on an electrical linear shaker for 4 min at 120 rpm at room temperature. The adsorbent was removed by filtration using a 0.45 µm filter using a syringe, and the residual or unadsorbed concentration of CTM was determined spectrophotometrically by measuring the absorbance at 238 nm using a linear regression equation derived from the calibration data obtained for different concentrations of CTM. The uptake of CTM by chemically modified adsorbent was calculated using the following equation:(16)$$q_{e}=\frac{\left(C_{i}-{\;C}_{e\;}\right)V}{m}$$
where q_e_ = amount of adsorbed citalopram HBr onto the adsorbent material (mg·g^−1^) at equilibrium time; V = solution volume (liter); m = amount of adsorbent (g). The % removal of CTM was calculated using the equation given below:(17)$$\mathrm{Removal}\left(\%\right)=\frac{C_{i}-{\;C}_{e}}{C_{i}}\times 100$$
where C_i_ and C_e_ = initial and equilibrium concentrations of citalopram (mg·L^−1^) at t = 0 and t = equilibrium time, respectively.

### 3.6. Desirability Approach

Derrigner’s desirability function [55] was utilized to convert each response (percent removal) into an individual desirability (df_i_) function. The scale of desirability function varied from 0 to 1, where zero indicates an undesirable response and one signifies the completely desirable response. For obtaining the maximum response (maximum percent removal), the value of df_i_ can be calculated by the following equation:(18)$$df_{i}=\left[\begin{array}{ccc} 0 & & \\ & {\left(\frac{R_{i}-L}{H-L}\right)}^{w_{i}} & \begin{array}{c} R_{i}<L\\ L\leq R_{i}\leq H\\ R_{i}\;>\;H \end{array}\\ 1 & & \end{array}\right]$$
where R_i_ = response value; H = highest value; L = lowest values. w_i_ defines the weight for the calculation of the desirability scale. After transforming the n variables into desirability functions, the individual functions are combined to obtain global desirability, D, as:(19)D=(df1×df2×…dfn)1n
where n = response number.

## 4. Conclusions

In this study, citric-acid-modified *Moringa peregrina* leaf adsorbent was utilized for the removal of citalopram HBr. The effect of pH, contact time, adsorbent dose and the initial concentration of citalopram HBr on the % removal of drug was investigated and optimized using a Box–Behnken plot. The quadratic equation developed for the % removal of citalopram HBr showed excellent correlation between the measured and predicted values. The optimized values at which the maximum percent removal (82.59%) was obtained were pH of 7, contact time of 5 min, adsorbent dose of 0.17 g and initial concentration of 35 mg·L^−1^. The adsorption kinetics followed pseudo second-order kinetic model and the diffusion-chemisorption model. The adsorption isotherm data fit well with the Freundlich model based on the least error obtained by chi-square. The Langmuir model was also followed based on R_L_ values greater than 0 and less than 1 with an appreciable coefficient of correlation. Hence, the adsorption isotherm data confirmed that the adsorbent surface was monolayer with some heterogeneous adsorption nature due to the chemisorption reactions as proved by the Freundlich model. The Gibb’s free energy change of −17.89 kJ·mol^−1^ was obtained at room temperature, which confirmed the feasibility and endothermic nature of the adsorption process.

## Figures and Tables

**Figure 1 pharmaceuticals-15-00760-f001:**
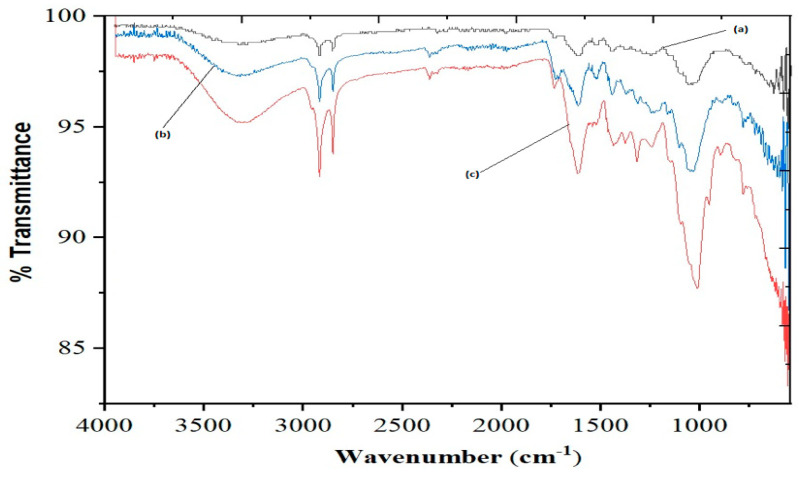
Infrared spectra of (**a**) *Moringa peregrina* biomass; (**b**) *Moringa peregrina* biomass treated with NaOH; (**c**) *Moringa peregrina* biomass treated with citric acid.

**Figure 2 pharmaceuticals-15-00760-f002:**
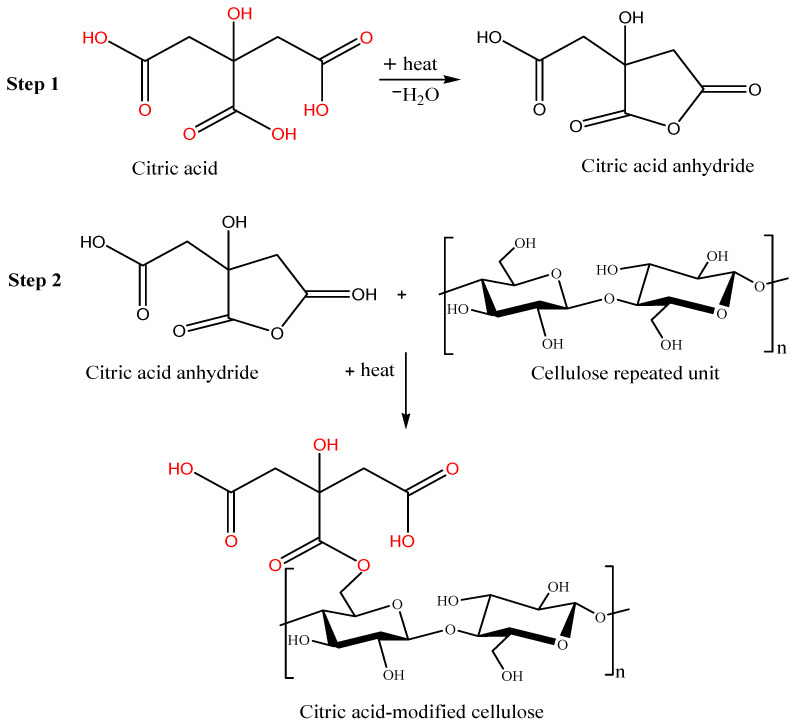
Reaction sequence of modified cellulose derived from *Moringa peregrina* with citric acid.

**Figure 3 pharmaceuticals-15-00760-f003:**
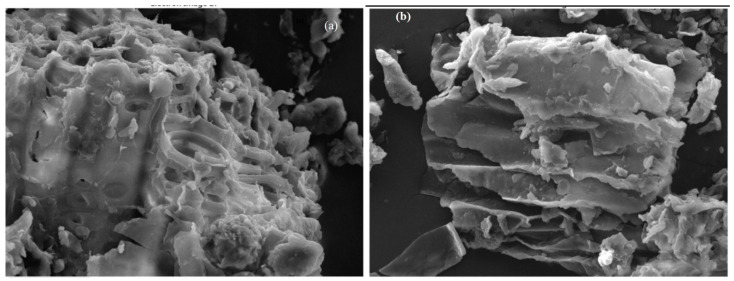
SEM images of (**a**) *Moringa peregrina* biomass and (**b**) *Moringa peregrina* biomass treated with citric acid.

**Figure 4 pharmaceuticals-15-00760-f004:**
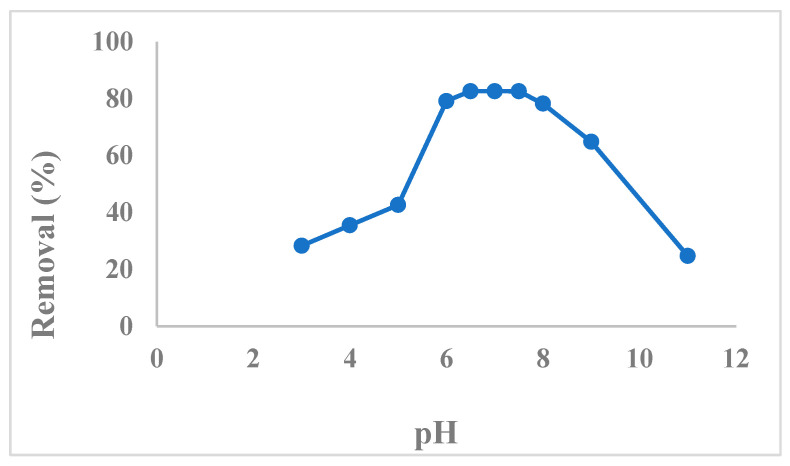
Effect of pH on the removal of CTM.

**Figure 5 pharmaceuticals-15-00760-f005:**
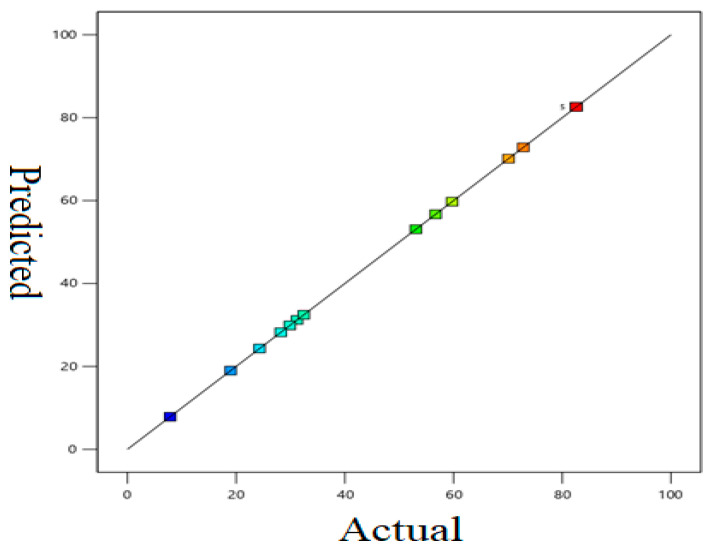
Plot of the experimental run in different colors refer to Table 1: % observed response versus % predicted response (blue color showed lowest and red color showed highest % removal of citalopram).

**Figure 6 pharmaceuticals-15-00760-f006:**
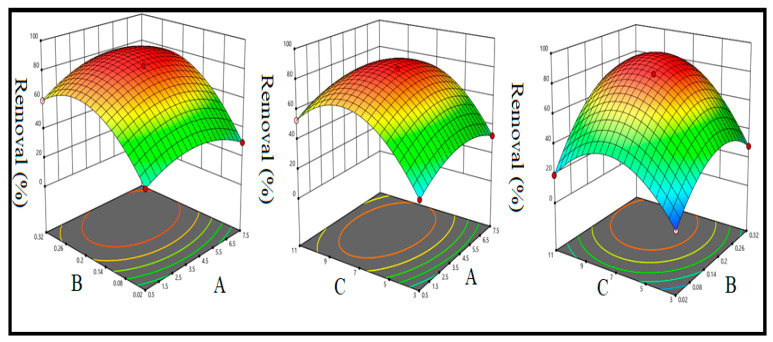
Response surface plots for the combined effects of (**A**) contact time and pH; (**B**) contact time and adsorbent dose; (**C**) pH and adsorbent dose on the % removal of citalopram.

**Figure 7 pharmaceuticals-15-00760-f007:**
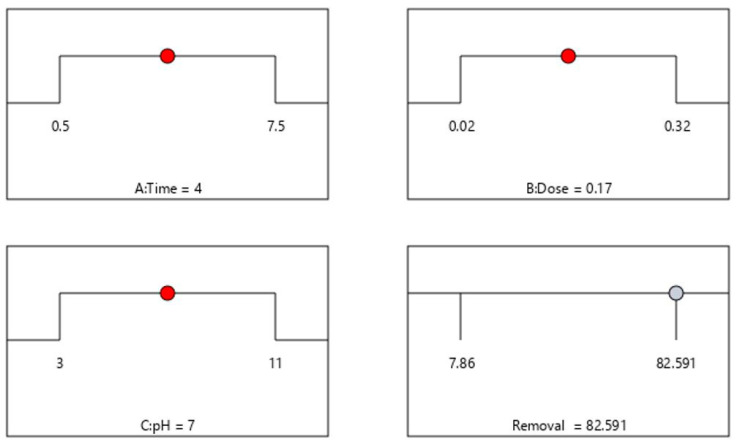
Desirability function-based profile: Optimized values of contact time (4 min), adsorbent dose (0.17 g), pH (7) and maximum % removal of CTM (82.59%).

**Figure 8 pharmaceuticals-15-00760-f008:**
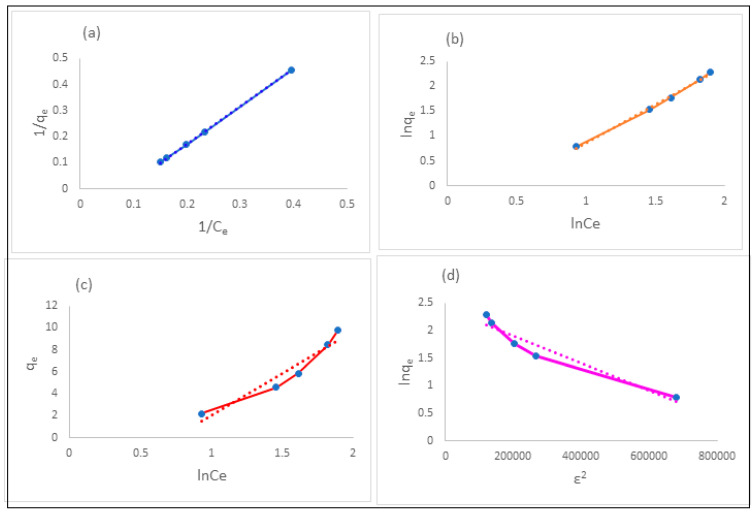
(**a**) Langmuir; (**b**) Freundlich; (**c**) Temkin; (**d**) Dubinin–Radushkevich adsorption isotherm plots for the adsorption of citalopram HBr onto citric-acid-modified *Moringa peregrina* leaf adsorbent at room temperature.

**Figure 9 pharmaceuticals-15-00760-f009:**
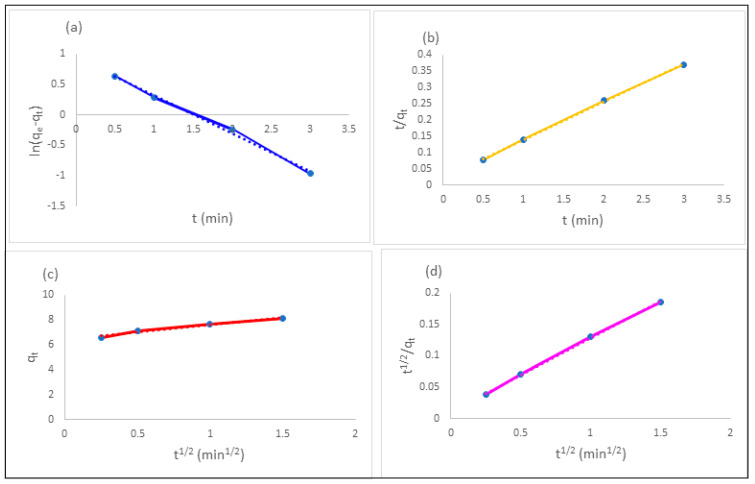
(**a**) Pseudo-first order; (**b**) pseudo-second order; (**c**) intraparticle diffusion; (**d**) diffusion-chemisorption plots for adsorption of CTM onto citric-acid-modified *Moringa peregrina* leaf adsorbent at room temperature.

**Table 1 pharmaceuticals-15-00760-t001:** Box–Behnken design matrix data: experimental and predicted values.

Experimental Run	A: Time (min)	B: pH	C: Adsorbent Dose (g)	Observed Response (%)	Predicted Response (%)
1	4	11	0.02	19.00	19.27
2	4	7	0.17	82.59	82.59
3	7.5	7	0.32	72.84	72.84
4	0.5	11	0.17	53.09	53.08
5	0.5	3	0.17	24.32	24.29
6	4	7	0.17	82.59	82.59
7	7.5	11	0.17	56.72	56.97
8	4	11	0.32	70.13	70.13
9	4	3	0.02	7.87	7.85
10	4	3	0.32	28.23	28.25
11	0.5	7	0.32	59.75	59.76
12	7.5	7	0.02	29.89	29.89
13	4	7	0.17	82.59	82.59
14	4	7	0.17	82.59	82.59
15	0.5	7	0.02	31.22	31.23
16	4	7	0.17	82.59	82.59
17	7.5	3	0.17	32.47	32.47

**Table 2 pharmaceuticals-15-00760-t002:** Statistical data of the polynomial models.

Model	SD	F	*p*-Values	R^2^	Adjusted R^2^	Predicted R^2^	PRESS
Linear	159.24	2.44	0.0189	0.2127	0.6767	0.4890	7.61 × 10^5^
Two-factor interaction	167.90	0.14	0.1450	0.0185	0.6405	−0.6754	2.450 × 10^6^
Quadratic	13.73	6.40 × 10^8^	<0.0001	1.0000	0.9999	0.9999	1951.71

**Table 3 pharmaceuticals-15-00760-t003:** Parameters of various adsorption isotherms calculated using linear equations for adsorption of CTM onto citric-acid-modified *Moringa peregrina* leaf adsorbent at room temperature.

Isotherm *	Parameters		R^2^	Error Function (χ^2^)
Langmuir	q_e_ (mg·g^−1^)	8.55	0.999	3.24 × 10^−3^
	K_L_ (L·mg^−1^)	0.08		
	R_L_	0.26		
Freundlich	q_e_ (mg·g^−1^)	8.41	0.996	3.77 × 10^−2^
	1/n	1.53		
	K_f_	0.51		
Temkin	q_e_ (mg·g^−1^)	8.30	0.919	6.54 × 10^−1^
	A_T_ (Lg^−1^)	0.49		
	B_T_ (J·mol^−1^)	7.56		
Dubinin–Radushkevich	q_e_ (mg·g^−1^)	8.20	0.926	1.04
	K_ad_ (mol^2^·J^−2^)	−2 × 10^−6^		
	E (kJ·mol^−1^)	−0.50		

* Initial concentration of CTM: 35 mg·L^−1^; adsorbent dose: 0.17 g; pH: 7; contact time: 4 min; rotation per minutes: 120.

**Table 4 pharmaceuticals-15-00760-t004:** Kinetic parameters for adsorption of CTM onto citric-acid-modified *Moringa peregrina* leaf adsorbent at room temperature.

Kinetic Model	Parameters		
Pseudo first order	q_e_ (mg·g^−1^)	k_1_ (min^−1^)	R^2^
	2.59	0.63	0.9975
Pseudo second order	q_e_ (mg·g^−1^)	k_2_ (g mg^−1^ min^−1^)	R^2^
	8.50	0.69	0.9995
Intraparticle diffusion	C_id_ (mg·g^−1^)	K_id_ (mg·g^−1^ min^−1/2^)	R^2^
	6.44	1.16	0.9810
Diffusion chemisorption	q_e_ (mg·g^−1^)	K_dc_ (mg·g^−1^ min^−1/2^)	R^2^
	8.50	98.04	0.9995

**Table 5 pharmaceuticals-15-00760-t005:** Comparison of the performance of citric-acid-modified cellulose adsorbent derived from *Moringa peregrina* leaf with other adsorbents for the removal of CTM.

Adsorbent	Contact Time (min)	Removal (%)	Adsorption Capacity (mg·g^−1^)	Reference
Paper mill sludge pyrolyzed at 800 ℃	30	-	8.50	[25]
SDS-coated magnetic particles	10	79.80	-	[26]
Reduced graphene oxide (rGO)	10	26.30	-	[27]
Nanoscale zero valent iron (nZVI)	10	31.41	-	[27]
rGO/nZVI	10	47.90	-	[27]
^#^ Porous alumina coated with natural zeolite	-	(i) 84.00 (ii) 75.00 (iii) 82.00	-	[28]
Citric-acid-modified cellulose adsorbent derived from *Moringa peregrina* leaf	4	82.59	8.58	This work

^#^ (i) Spiked water; (ii) spiked surface water; (iii) spiked wastewater treatment plant effluent.

**Table 6 pharmaceuticals-15-00760-t006:** Independent variables and their levels used for the Box–Behnken design.

Variables	Unit	Factor	Range and Level		
			−1	0	+1
Contact time	minute	A	0.50	4.00	7.50
pH		B	3.00	7.00	11.00
Adsorbent dose	g	C	0.02	0.17	0.32

## Data Availability

Data is contained within the article.

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
