# Peer review of "Development of a Citric-Acid-Modified Cellulose Adsorbent Derived from Moringa peregrina Leaf for Adsorptive Removal of Citalopram HBr in Aqueous Solutions"

_pharmaceuticals, 2022, doi:10.3390/ph15060760_

Round 1

Reviewer 1 Report

This manuscript deals with preparing a Citric acid-modified Moringa peregrina leaf adsorbent for citalopram HBr (CTM) adsorption. The authors used Box Behnken design to determine the optimum conditions, and they studied one response %R. The produced adsorbent was characterized using the IR spectrum and showed the presence of significant absorption bands. The equilibrium studies showed that the maximum adsorption capacity was 8.58 mg/g. In my opinion, this work is incomplete, and significant issues require careful revision before publication. Specifically:

  1. Section 2. Results:
    • The adsorbent characterization: IR analysis is not enough to study the effect of modification on the studied adsorbent. The authors need to do other characterization techniques like CHN analysis, Raman, TGA, BET, and SEM analysis
    • Figure 1: the IR analysis looks not good to me, and it should be drawn using origin lab or excel.
    • Line 149: Optimization of variables: The author prepared 3 samples and studied only one sample, I think they should perform the same design with the other two samples and compare them to show the effect of the modification process.
    • Table 1: The highest %R was found to be 82.59%, which is very low compared to other studies.
    • Line 206: Adsorption isotherm. The authors should put all the obtained parameters in a table, and they should be clearly identified. They should also add the figures of the equilibrium isotherms. The authors did not show the type of equilibrium isotherm, whether linear or nonlinear isotherms.
    • Line 216: The qm obtained from the Langmuir isotherm was found to be 8.58 mg/g, which is very small, and I think the authors need to do further modification or convert their adsorbent to biochar to increase the surface area and could improve the adsorption capacity.

Author Response

RESPONSE TO REVIEWER No.1 COMMENTS

REVIEWER # 1

Comment # 1: Section 2. Results: The adsorbent characterization: IR analysis is not enough to study the effect of modification on the studied adsorbent. The authors need to do other characterization techniques like CHN analysis, Raman, TGA, BET, and SEM analysis.

Response: SEM analysis is included. Other characterization techniques have not been carried out due to short time given for revision.

Comment # 2:  Figure 1: the IR analysis looks not good to me, and it should be drawn using origin lab or excel.

Response: Improved Figure 1 is added.

Comment # 3: Line 149: Optimization of variables: The author prepared 3samples and studied only one sample, I think they should perform the same design with the other two samples and compare them to show the effect of the modification process.

Response: Moringa peregrina leaf biomass, Moringa peregrina leaf biomass treated with NaOH and citric acid have shown the removal efficiency of 59.45%, 71.36% and 82.59%, respectively. Therefore, Moringa peregrina leaf biomass treated with citric acid was selected for further studies.

Comment # 4: Table 1: The highest %R was found to be 82.59%, which is very low compared to other studies.

Response: The highest percent removal was found to be 82.59%, but tis removal was obtained in 4 min only.

Comment # 5: Line 206: Adsorption isotherm. The authors should put all the obtained parameters in a table, and they should be clearly identified. They should also add the figures of the equilibrium isotherms. The authors did not show the type of equilibrium isotherm, whether linear or nonlinear isotherms.

Response: Adsorption isotherm parameters are given in Table  3  . Equilibrium isotherms are presented in Figure8 (a-d).Linear isotherms were discussed.

Comment # 6: Line 216: The q obtained from the Langmuir isotherm was found to be 8.58 mg/g, which is very small, and I think the authors need to do further modification or convert their adsorbent to biochar to increase the surface area and could improve the adsorption capacity.

Response: This suggestion is very nice and I shall prepare biochar for further studies.

Reviewer 2 Report

Review for pharmaceuticals-1708797

My review is centered around a scanned pdf of the manuscript that includes both linguistic suggestions and comments/question marks.

The detailed comments pertain to numbered items in that manuscript (#i) to which I will refer in order of appearance.

#1: Are two digits significant? What is the accuracy/precision of your measurements? The model should not try to be more accurate than that.

#2 (lines 30,19): the units in mg/g are not making much sense (see general remark C below). First the reader does not know what CTM is (i.e. how big it is) and then the adsorbed amount (preferably also in moles) should be referenced to the surface area, since we deal with a surface process. Ideally you give both, since from an operative point of view the per mass could be interesting, but from a fundamental point of view the area is required.

#3 the time is confusing. If in 1994 the review occurred, then it cannot be considered insignificant till the end of the nineties.

#4 I suggest to consistently use the abbreviation CTM, and it has to be clear whether citalopram and citalopram HBr is the same or if there are differences.

#5 explain the abbreviation RSM

#6 Again the digits are difficult to understand. I guess omitting them would be good here for the wavenumbers. Not sure they make much sense on the graphs with the spectra.

#7 same as with #1. And in the same kind of thought, I do not recall having seen how accurate/precise the measurement of CTM is with the method used here. And did you do replicates?

#8 same as #7 and in this case I was wondering about how precise you can keep pH 7. This is usually the most difficult to keep pH, if there is no buffering. Was carbon dioxide excluded in the measurements? Please state these conditions precisely. How was the pH adjusted? i.e. which acid/base solutions were used at which concentrations? If the pH is somewhat varying during the isotherm measurements or adsorption measurements in general, then it is part of the variation, since you have a pH-dependent process. A pH-edge/envelope would be nice (i.e. a curve of the uptake as a function of pH with more than three values). What is known about CTM speciation in solution (i.e. pH dependence)?

#9 the ability maybe? In general for this kind of study and all the hundreds similar ones, I do not understand why one still resorts to these kind of conditions isotherms when it is clear that you have a pH-dependent process. A reasonable study also should involve the study of the surface properties of the sorbent as a function of pH (like determining the point of zero-charge, measuring zeta-potentials or doing potentiometric titrations) and then using a more mechanistic model. It has to be clearly stated here also which pH the isotherm etc. corresponds to.

#10 this is the linearized form (or one of them). Using another linearization would produce different parameters. Was this tested? Please show on a graph the isotherm data and the different fits. The symbol qe does not seem to be defined here.

#11 Again the question about significant digits throughout this section…

#12 the two numbers should be the same, right? Or is the 1.459 from the literature reference. Then it would still be confusing.

#13 I guess the chi-square and the correlation coefficient discussion is somewhat redundant, if you deal with linarizations, right?

#14 again digits. I do not believe you can fix (or even measure) pH to the second digit in a suspension that does not contain a background electrolyte (which by the way should be done, because otherwise you might be changing the activity coefficients on the isotherm range and then the treatment is getting inconsistent).

#15 This is now a linear isotherm, or a kind of Henry coefficient? If you have Langmuir behavior then you would have data outside the linear range and can get into trouble. Without showing the data, this cannot be judged at all.

#16 not clear either. The qe is from Langmuir isotherm evaluation I gather? Please make clear it all is the same pH. What happens if you do the kinetic experiment at another concentration or solid content? Do you get another set of parameters?

#17 It is nice they co-incide like this, but I doubt it will still work if you do kinetics at more than one solid concentration or total CTM concentration. This is too easy.

#18 for this you had to involve acid or base addition. Is CTM anionic or cationic? During the adsorption, usually protons are co-adsorbed or released and if you want to work at constant pH, you need to adjust this continuously. Is this what was done? Or was just the initial pH adjusted? Also say how you calibrated your electrode and what electrode you used. As I pointed out these kind of experiments are best done with a constant background electrolyte.

Finally I have the following general comments:

  1. Most journals I read (if not all) on a regular basis have the experimental design before the results and discussion section. This allows to understand what has been done before the results are presented. Maybe this journal has a specific order that deviates from what I prefer. But I would encourage this sequence.
  2. I would encourage to number the equations
  3. One information that was lacking for me is the specific surface area of the material. Since we deal with adsorption, we expect that the surface is the most important issue, not the mass of the adsorbent. As pointed out in the remarks above, the comparison of uptake makes sense mostly on the per surface area basis. It also gives a feeling about the coverage, since there are many numbers in µmoles/m2 for example available in the literature so that numbers in those units would indicate if the measurements are in a reasonable range. Moreover this can be easily transferred and compared to surface site densities (to which e.g. the Langmuir model ultimately pertains).

Overall I see a lot of missing details. The authors should revise. I would not recommend publication, and in general think that these kind of studies should not be done on the very conditional basis. The state of the art is much more advanced, and the conditional parameters, studies are in the end useless for advancing science.

Author Response

RESPONSE TO REVIEWER No. 2 COMMENTS

Reviewer # 2

Comment # 1: Are two digits significant? What is the accuracy/precision of your measurements? The model should not try to be more accurate than that.

Response: The measurement was made up to two places after decimal.

Comment # 2: (lines 30,19): the units in mg/g are not making much sense (see general remark C below). First the reader does not know what CTM is (i.e. how big it is) and then the adsorbed amount (preferably also in moles) should be referenced to the surface area, since we deal with a surface process. Ideally you give both since from an operative point of view the per mass could be interesting, but from a fundamental point of view the area is required.

Response: In most of (more than 98%) studied capacity is expressed in mg/g. So I think it is acceptable practice.

Comment # 3: the time is confusing. If in 1994 the review occurred, then it cannot be considered insignificant till the end of the nineties.

Response: It was corrected.

Comment # 4: I suggest to consistently use the abbreviation CTM, and it has to be clear whether citalopram and citalopram HBr is the same or if there are differences.

Response: CTM was used throughout the text.

Comment # 5: explain the abbreviation RSM

Response: Response surface methodology—it is explained in the manuscript.

Comment # 6: Again the digits are difficult to understand. I guess omitting them would be good here for the wavenumbers. Not sure they make much sense on the graphs with the spectra.

Response: Omitted.

Comment # 7: same as with #1. And in the same kind of thought, I do not recall having seen how accurate/precise the measurement of CTM is with the method used here. And did you do replicates?

Response: three replicates have been done and the average value is reported.

Comment # 8: same as #7 and in this case I was wondering about how precise you can keep pH 7. This is usually the most difficult to keep pH, if there is no buffering. Was carbon dioxide excluded in the measurements? Please state these conditions precisely. How was the pH adjusted? i.e. which acid/base solutions were used at which concentrations? If the pH is somewhat varying during the isotherm measurements or adsorption measurements in general, then it is part of the variation, since you have a pH-dependent process. A pH-edge/envelope would be nice (i.e. a curve of the uptake as afunction of pH with more than three values). What is known about CTM speciation in solution (i.e. pH dependence)?

Response: Solution pH was maintained at 7 with 0.1 M HCl or 0.1 M NaOH.

Comment # 9: the ability maybe? In general for this kind of study and all the hundreds similar ones, I do not understand why one still resorts to these kind of conditions isotherms when it is clear that you have a pH-dependent process. A reasonable study also should involve the study of the surface properties of the sorbent as a function of pH (like determining the point of zero-charge, measuring zeta-potentials or doing potentiometric titrations) and then using a more mechanistic model. It has to be clearly stated here also which pH the isotherm etc. corresponds to.

Response: Effect of pH on the adsorption was studied. Point of zero charge was also determined and found to be 4.8.

 Comment #10: this is the linearized form (or one of them). Using another linearization would produce different parameters. Was this tested? Please show on a graph the isotherm data and the different fits. The symbol qe does not seem to be defined here.

Response: Isotherm plots are shown in Fig 8(a-d). and symbol qe is defined in the manuscript.

Comment # 11: Again the question about significant digits throughout this section…

Response: Corrected.

Comment # 12: the two numbers should be the same, right? Or is the 1.459from the literature reference. Then it would still be confusing.

Response: Corrected.

Comment # 13: I guess the chi-square and the correlation coefficient discussion is somewhat redundant, if you deal with linearizations, right?

Response: Corrected

Comment # 14: again digits. I do not believe you can fix (or even measure) pH to the second digit in a suspension that does not contain a background electrolyte (which by the way should be done, because otherwise you might be changing the activity coefficients on the isotherm range and then the treatment is getting inconsistent).

Response: corrected.

Comment # 15: This is now a linear isotherm, or a kind of Henry coefficient? If you have Langmuir behavior then you would have data outside the linear range and can get into trouble. Without showing the data, this cannot be judged at all.

Response: Data are shown in Table 4 & Fig 8(a-d)

Comment #16: not clear either. The qe is from Langmuir isotherm evaluation Igather? Please make clear it all is the same pH. What happens if you do the kinetic experiment at another concentration or solid content? Do you get another set of parameters?

Response: qe is experimental value and qmis obtained from Langmuir plot.

Comment # 17: It is nice they co-incide like this, but I doubt it will still work if you do kinetics at more than one solid concentration or total CTM concentration. This is too easy.

Response: Kinetic studies were performed using 35 mg/L CTM solution.

Comment # 18: for this you had to involve acid or base addition. Is CTM anionic or cationic? During the adsorption, usually protons are co-adsorbed or released and if you want to work at constant pH, you need to adjust this continuously. Is this what was done? Or was just the initial pH adjusted? Also say how you calibrated your electrode and what electrode you used. As I pointed out these kind of experiments are best done with a constant background electrolyte.

Response:  CTM exists as cation in the pH range 5-9 and it is discussed under ‘Effect of pH on adsorption’.

General Comments

#1: Most journals I read (if not all) on a regular basis have the experimental design before the results and discussion section. This allows to understand what has been done before the results are presented. Maybe this journal has a specific order that deviates from what I prefer. But I would encourage this sequence.

Response:

# 2: I would encourage to number the equations

Response: Corrected

# 3: One information that was lacking for me is the specific surface area of the material. Since we deal with adsorption, we expect that the surface is the most important issue, not the mass of the adsorbent. As pointed out in the remarks above, the comparison of uptake makes sense mostly on the per surface area basis. It also gives a feeling about the coverage, since there are many numbers in μmoles/m2 for example available in the literature so that numbers in those units would indicate if the measurements are in a reasonable range. Moreover this can be easily transferred and compared to surface site densities (to which e.g. the Langmuir model ultimately pertains).

Response: Due to short time given for revision, it seems to be not possible due to heavy rush on Instrumental facilities centers.

Reviewer 3 Report

This paper presents an interesting strategy for searching for the synthesized of chemically modified Moringa peregrina leaf adsorbent and then applying it to the adsorption of citalopram HBr in aqueous solutions by batch adsorption methods. Furthermore, isotherm models, kinetic and thermodynamic studies were performed. Although considerable work has been performed, several points must be improved for the acceptance of this manuscript.

  1. The resolution of Figure 1 is low. Please improve it.
  1. The equations have to be numbered as they first appear in the text (ex. (1) in page 10 line 212)
  1. Be careful at subscript Ce in page 10 line 212.
  1. The last part of the introduction section must be rewritten, page 3 lines 91-97. The aim of study must be better presented.
  1. Why did you choose only 3 pH values? I recommend you investigate the 5 and 9 values as well.
  1. Please compare the adsorption capacities with the ones presented in the research literature for similar adsorbents.
  1. The references must be abbreviated as written in the instructions: https://www.mdpi.com/journal/pharmaceuticals/instructions

Based on these, I advise the authors to rectify the above mentioned errors, and I hope to re-evaluate the revised manuscript.

Author Response

RESPONSE TO REVIEWER No. 3 COMMENTS

REVIEWER # 3

Comment #1: The resolution of Figure 1 is low. Please improve it.

Response: Improved Figure 1 is included in the manuscript.

Comment #2: The equations have to be numbered as they first appear in the text (ex. (1) in page 10 line 212)

Response: Corrected.

Comment # 3: Be careful at subscript Ce in page 10 line 212. 

Response: Corrected

Comment # 4:The last part of the introduction section must be rewritten, page3 lines 91-97. The aim of study must be better presented.

Response: corrected

Comment # 5: Why did you choose only 3 pH values? I recommend you investigate the 5 and 9 values as well.

Response: Effect of pH(including pH 5 & 9) has been added.

Comment # 6: Please compare the adsorption capacities with the ones presented in the research literature for similar adsorbents.

Response: itb is given in introduction section.

 Comment # 7: The references must be abbreviated as written in the instructions:https://www.mdpi.com/journal/pharmaceuticals/instructions

Response: corrected.

Round 2

Reviewer 1 Report

The paper improved, and it can be published in its present form.

Author Response

Comment: The paper improved, and it can be published in its present form.

Response: Grateful to reviewer. Thank you very for your specialized comments and acceptance of the revised manuscript.

Reviewer 2 Report

Review for Azmi et al.

I continue to have some serious reservations about this manuscript. As in the previous round my review is centered around a scanned pdf of the manuscript that includes both linguistic suggestions and comments/question marks.

The detailed comments pertain to numbered items in that manuscript (#i) to which I will refer in order of appearance.

#1 were they selected or obtained by the fitting?

#2 I strongly encourage to include a discussion on how the second significant digit on the percentage adsorbed can come about. It is related to the experimental errors. I have hardly ever come across the fourth significant number in a classical adsorption experiment like those reported here. It can happen with radioactive traces where you can control all kinds of losses. But for measurements of this kind, it is unacceptable, the more so, if the authors do not give details about the experimental analytical technique…

#3 there is always electrostatic interaction (except maybe at the point of zero charge/isoelectric point). Thus here it should be attraction.

#4 it is not clear why it should be maximum a that pH range. Probably the authors infer it from the experimental data. If the solute bears a positive charge over the whole pH-range, the attraction should increase with increasing pH, because the surface gets ever more negative. In other words, the speciation as a function of pH of the solute would be very useful here.

#5 Not clear why the hydroxide ions should surpass electrostatic interactions? Could be that the solute is taking up a hydroxide ion, but then you will have to rephrase this. Again the speciation plot would be very useful.

#6 if I count the number of adjustable parameters in eq. 1 and compare the 10 adjustable to the number of data points used in the fitting (17), I am not surprised you get a perfect fit. I have a feeling that you omit at least part of the data in Figure 4 (omit dot after Figure please), which could be predicted by the model, if I understand correctly. In the caption to Figure 4, you will need to give the amount of sorbent per solution volume, the CTM concentration (range, i.e. whatever is applicable) and the time. You can also cut the pH range at pH 2.

#7 Please pay close attention to units of your parameters. I marked it sometimes on the scanned copy as well. In table 3, I think for the Langmuir isotherm the parameter should not be qe but qmax. For the Freundlich isotherm you cannot obtained a limiting concentration. Please correct these things. The Freundlich constant has units, which are not given here. Again you have the problem of the number of data points (5 on the experimental isotherm) and the fit-parameters (3 in the Langmuir model). This is not good. Of course you will get a good fit.

#8 the parameters in those equations should have units.

#9 a thickness should have units of [m] or something like that; please use other terms…

#M1 this is not completely claear.

#M2 the blender mentioned above? Please clarify.

#M3 in step 2, how was the neutrilzation carried out.

#M3 (in step3) same as before.

Reviewer 3 Report

The author has made substantial improvements to this article. The manuscript can be accepted for publication after performing the below mentioned issue.

Please add a table to the end of the manuscript to compare the adsorption capacities with the ones presented in the research literature for similar adsorbents. It is not easy to follow the data presented in the introduction section.

Author Response

Comments: The author has made substantial improvements to this article. The manuscript can be accepted for publication after performing the below mentioned issue.

Please add a table to the end of the manuscript to compare the adsorption capacities with the ones presented in the research literature for similar adsorbents. It is not easy to follow the data presented in the introduction section.

Response: 

I am very grateful to reviewer for his valuable comments.

As per reviewer's comments, Table 5 is added in the revised manuscript to compare the performance of the adsorbent used in this study with other adsorbents and discussion is presented in Section 2.7 (Comparison with other adsorbents).